mechanical engineering/materials science

frugal, failure, adaptation, complex system, sustainable development, electric vehicle

**Author for correspondence:**
Balkrishna C. Rao
e-mail: balkrish@iitm.ac.in

# On complex systems of adaptive frugal products

## Balkrishna C. Rao

Department of Engineering Design, Indian Institute of Technology Madras, Chennai 600036, Tamil Nadu, India

BCR, 0000-0002-5984-8538

*Frugal* products possess a proper mix of features including minimal consumption of resources, good functionality under nominal conditions and low cost. Therefore, increasing use of *frugal* products, that are designed and also fabricated systematically, is crucial to all-round *sustainable development*. However, their low factor-of-safety rigorous-design makes them inherently prone to failure under conditions of overloading. And multitudes of such coupled-products would create topologies of interconnected complex systems in the foreseeable future whose individual products should be made to adapt against any events of failure to enhance functionality while maintaining low cost. Accordingly, this paper proposes a two-pronged methodology for adaptation of *frugal* products along with ramifications of complex systems of *frugal* products. The adaptation methodology is crucial to the functioning of individual and also networks of *frugal* products and this work accordingly explicates scenarios of ensuing networks. Other than application to various sectors including *electric vehicles*, a basic example of which is covered in this paper, the proposed adaptation-and-networking framework can also be applied to a growing numbers of sustainable products, which are *frugal* according to the terminology of this effort and hence prone to premature failure.

## 1. Introduction

In recent years, products of the *frugal* type have attained prominence and diffused around the world [1–4] due to their potential to improve living standards of society at large. This is due to the affordability of quality products realized through restrictions on consumption of resources for producing no-frills versions of electrocardiograms, cars and plasma technology, with more examples covered in [4,5]. The examples showcased by Rao [4,5] cover *frugal* products of both the *grassroots* and *advanced* types in sectors including healthcare, automotive, astronomy and particle physics, to name a few. When compared with conservative designs that employ material padding for enhancing safety [6,7], the no-frills nature of *frugal* products leads typically to less

extraction of material resources and also limits emissions of greenhouse gases (GHG) [7]. Therefore, low cost, lower material wastage in design and fabrication, and good functionality make *frugal* products an appealing sustainable alternative to incumbents. It should be noted that *frugality* through non-material resources, such as supply chains and sales, for lowering costs are not pursued in this effort.

The absence of excess material padding—i.e. typical of conservative designs [7]—typically results in lower factors of safety for *frugal* products [8,9]. Hence, rigorous design procedures have to be followed for achieving robust performance while accommodating the resulting lower factors of safety. The rationale for assuming that companies such as GE® and Tata Sons®, that have created the electrocardiogram and Nano-Car, respectively, have undertaken rigorous design procedures to create *frugal* products is that the underlying sophistication in technology requires one. Besides, *frugal* products can be designed from scratch through a rigorous design methodology outlined by Rao [8].

Further fail-proofing of *frugal* devices can be achieved by connecting these products in a network through the Internet of things (IoT) to enable 'talk' benefitting their performance. The advent of interconnectivity between products, including *frugal* ones, is inevitable considering the advancement in IoT and other technologies [10,11]. The 'talk' in a network would forewarn of the arrival of failure conditions that could be used for enhancing performance or discarding affected *frugal* products, depending on the context. In fact, network 'talk' would even buttress the aim of achieving low cost because a cheap no-frills structure can be complemented by information from network that would help maintain product functionality under adverse conditions.

Moreover, interconnectedness between ever-increasing numbers of *frugal* products to achieve a goal would create a complex system because of the high likelihood of failure of some *frugal* products, possibly due to perturbation from marginal overloads, and the subsequent functioning necessary of the underlying complex network through dynamic evolution of network topology. So any attempt at making *frugal* products adaptive would greatly aid in achieving controllability of networks associated with complex systems, thereby minimizing propagation of failures. In particular, adaptation would involve either product termination or morphing to strengthen for survival, in the face of adversity, thereby making a network of such products also adaptive, which by definition is complex [12].

Accordingly, sections of this paper have been organized to present the rationale underlying the complex-system angle on *frugal* products and the consequent need for making individual products adaptive. Subsequently, adaptation theory with a simple example to highlight the proposed concept is presented for futuristic employment in real-time applications. Lastly, local and global networks of *frugal* products have been described to facilitate futuristic utilization with a rudimentary example involving *electric vehicles* (EV). It should be noted that this work sets a framework and is therefore general in its scope since both adaptation methodology and the complex-systems angle on *frugal* products have not been reported to date.

## 2. Frugal products and complex systems

*Frugal* products are built against constrained use of resources and as such are typically vulnerable to failures under nominal conditions and/or conditions of even marginal overloading [9]. Hence a plausible solution is adaptation against impending failures involving product fortification or termination. It should be noted that this effort uses the term 'fortify' instead of 'strengthen' due to the latter's connotation of a material's yield or tensile strength which will be used in an example covered in this effort. If a variety of adaptive *frugal* products are integrated into a network to achieve a goal, there is still a significant chance of a certain proportion failing, and hence such a network should function, all the time, with some failed members and network failure happening when proportion of failed products goes beyond a certain threshold [13]. Therefore, interconnectivity between increasing numbers of different varieties of adaptive *frugal* products, for achieving a goal, would constitute a heterogeneous complex network [14], underlying a complex system, wherein failure of some products would not alter the functionality of the network [15]. Moreover, as a network of adaptive *frugal* products grows with time through natural evolution, as opposed to being systematically engineered, it will tend to being more complex.

The design and adaptation of *frugal* devices have a bearing on dynamics of both individual products and their interactions, and hence influence controllability of the underlying network [16]. In the event of adversity, adaptation would aid in retaining functionality of at least critical *frugal* products and thereby obstruct the cascading of failure. Moreover, affordability of complex systems in general [17] lends support to developing low-cost *frugal* products that can adapt against impending failures, and which subsequently are networked, for achieving a goal. In so doing, adaptation will lower, but not avoid

altogether, the redundancy of products in the network. Hence, it is imperative to impart adaptive abilities to individual *frugal* products for facilitating robust functioning of network under real-time conditions; such adaptation—involving fortification or rejection of concerned *frugal* product(s)—being based on identification of weak spots through theory and their subsequent continual monitoring for impending failures through suitable sensor networks involving IoT.

## 2.1. Linking frugal products

*Frugal* systems can span either a local or global network, wherein the former refers to isolated products such as an automotive vehicle or a mobile phone and the latter denotes larger interconnectedness between local networks or products. For both local and global networks, the design and adaptation of their constituents is crucial for effective functioning of system as a whole and also for generating complex behaviour. In fact, adaptation of individual constituents and also interactions between them will remain invariant under various perturbations and/or network topologies, especially for global networks [18].

In recent years, advent of the IoT is enabling wireless communication between a variety of devices equipped with controllers, means for wireless communication, protocols etc. for mutual interaction and also 'talk' with users in completing various tasks to fruition. In so doing the IoT will facilitate integration of various devices with the Internet [19,20]. Therefore, other than standard couplings, such as mechanical ones between *frugal* products, IoT should also be used, as applied in this work, to enhance connectivity and also abet product adaptation through features of IoT-enabled devices to provide for appearance of complex behaviour in networks, local or global.

# 3. Methodology for adaptation

Rigorous procedures have to be employed for designing *frugal* products due to constraint on materials consumed during their realization. In particular, lesser amounts of raw materials usually translate into lower *factors of safety*, in *classical* design, that increase the likelihood of failure arising under loading that is outside the range of safety [9]. Examples of failure due to overloading abound in various sectors such as railroad [21,22], aerospace [23], shipping [24] and medical [25], to name a few. Consequently, this effort focuses on failures due to overloading, while risks posed by nominal conditions through material defects, product geometry and other product-based factors [26] controlled by rigorous design in the *factor of frugality* approach by Rao [8,9].

Rigorous design procedures are imperative for *frugal* products, whose exclusion can lead to failures under even nominal conditions and which might be the case for *grassroots frugal innovations* (GFIs) which are makeshift contraptions. Even though some overloading could be sustained by a rigorously designed *frugal* product with a low *safety factor* [9], approximately 1.5, this effort assumes zero tolerance for overloading. Therefore, sensor(s) should be embedded in fortified *frugal* products around their weakest areas for continuous health-monitoring so as to take remedial action when abnormal conditions, in the form of excess loading, arrive. Sensor signals from a network of fortified products will facilitate forewarning of failure on a target product that is away from the approaching failure. Consequently, two courses of action are available when conditions of failure are sensed by a frugal product. First, the product could be triggered to terminate functioning when there is a swarm of products collectively working to achieve a goal. Second, the weakest areas of the product can be strengthened by additional features that are instantly initiated by forewarning from sensors. In other words, sensors will abet either termination of a concerned fortified *frugal* product altogether or initiate a remedial action involving possibly morphing to withstand excess load. All of this is possible under the assumption of availability of a window of time from receiving signal to the undertaking of actual action.

Therefore, this work presents a two-pronged approach for adaptation of individual *frugal* products wherein 'fortifying' a product through rigorous design of the *factor of frugality* approach [8,9] is followed by 'forestalling' of impending failure by *health monitoring* of critical areas through embedded sensors and subsequent remedial action. This adaptation methodology would henceforth be referred to as *fortification and forestalling* (FAF).

An existing case study on the design of a shaft [27] has been selected for exemplifying the proposed FAF methodology of adaptation. The focus on shaft is justified by its significance as one of the basic blocks in engineering design, widely used to transmit power and/or conversion between linear and rotary motions.

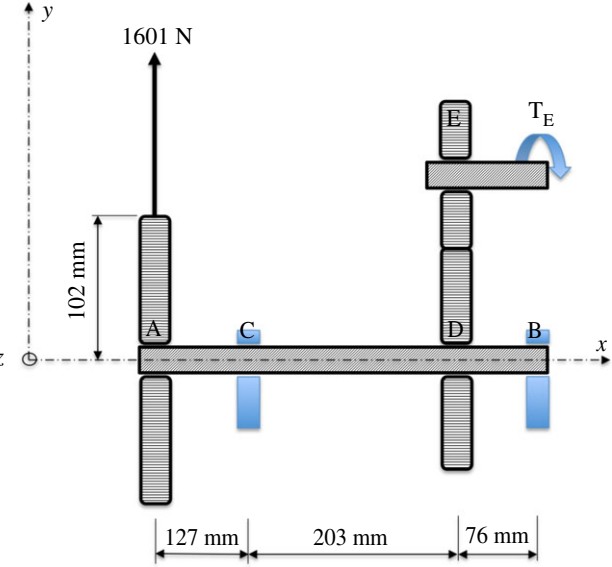

**Figure 1.** Shaft under torsional and bending loads [27].

**Table 1.** Results of baseline shaft design.

| s. no. | factor of safety (S) | material | critical section | diameter of critical section (D mm) |
|---|---|---|---|---|
| 1 | 1.6 | AISI 1010CD | C | 24.2 |

**Table 2.** *Factor of frugality* for the shaft.

| s. no. | factor of safety (S) | material saved | | | factor of frugality (F$^S$) |
| | | simple design (MS$_1$) | manufacturing (MS$_2$) | 4R mechanisms (MS$_3$) | |
|---|---|---|---|---|---|
| 1 | 1.6 | 0 | 0 | 0 | 1.6 |
| 2 | 1.6 | 0.5 | 0 | 0 | 2.1 |
| 3 | 1.6 | 0.5 | 1 | 0 | 3.1 |
| 4 | 1.6 | 0.5 | 1 | 1 | 4.1 |

## 3.1. Fortification through factor of frugality

Figure 1 illustrates a shaft to be designed and also fabricated for *frugality* against torsional and bending loads acting on it. Gear A carries a load of 1601 N and is attached to shaft AB, made of AISI 1010 steel, whose other end is connected to gears D and E. The details of designing shaft AB by the *classical factor of safety* approach is given in Ugural [27], where a *safety-factor* (S) of 1.6 has been applied to get the results listed in table 1.

The results pertaining to *factor of frugality* (F$^S$) for shaft AB are listed in table 2, where the methodology outlined by Rao [8] has been applied to the baseline data of table 1. The columns grouped under *material saved* (MS) contain scores for various extraneous material-saving schemes quantified by Rao [8]. Each entry of table 2 highlights contribution of a scheme to conserving resources with baseline design of table 1 listed as first. Here, F is equated to S since no extraneous scheme is used in baseline design to save material other than that resulting from a design with low *factor of safety*. The remaining entries of table 2 outline savings from individual extraneous material-saving schemes except for the last entry, which also lists cumulative F number aggregated from S and MS values corresponding to all schemes.

The second entry of table 2 quantifies material saved by replacing a solid shaft with a hollow one through $MS_1$ of *simple design* scheme using the maximum shear stress theory of failure [27]. The third entry pertains to a scheme that uses extrusion for minimizing wastage of excess material removed during manufacturing. Accordingly, $MS_2$ is assigned one to reflect avoidance of having to machine the internal diameter of a solid shaft to arrive at the hollow feature. And entry 4 relates to salvaging of such an extruded hollow shaft from a suitable *end of life* (EOL) system. The parameter $MS_3$, representing *4R mechanisms* scheme for raw material conservation through salvaging, is assigned a value of one assuming that the hollow shaft has been procured from a discarded product and not compromised in functionality. The three schemes taken together with S result in a F value of 4.1 which shows an improvement of 156% vis-à-vis the F or S value of 1.6 for the baseline design. Therefore, the final design consists of an extruded hollow shaft, which will be procured from an EOL system, whose weakest section, shown in figure 1, is at C.

## 3.2. Forestalling by shaft

The weakest section C should be inspected in real time by suitable structural-health-monitoring techniques [28] for forestalling failure. The shaft on notification of arrival of excess loading could change configuration at C using morphing techniques with typical representative examples reported in [29,30]. Morphing in general would aid development of self-healing *frugal* products [31]. Alternatively, the shaft could shut itself, without recourse to healing, thereby deactivating several links possibly connected to it.

The FAF methodology can also be applied to other components such as, and not limited to, plates, beams etc. It should be noted that the shaft is a representative *frugal* product, and generally *frugal* devices have to be designed and adapted according to the FAF methodology prescribed here. Accordingly, an adaptive *frugal* product could be used in an isolated application or could be part of a network of only *frugal* products underlying possibly a complex system [31]. Although FAF could be applied to both applications, this work focuses on networks associated with *frugal* products.

# 4. Frugal products and networks

This section describes two types of networks associated with *frugal* products that could materialize in the foreseeable future. Accordingly, *frugal*-product-as-a-network-of-*frugal*-components and networks-of-*frugal*-products will facilitate understanding of changes at individual product level on the system as a whole. An assumption implicit in this effort is that the price of any *frugal* product will be low due to their eventual commercial success.

## 4.1. Frugal product as network of frugal components

An individual assembly of *frugal* components is a stand-alone local network forming an isolated *frugal* product. Examples include *frugal* versions of arbitrary products including computers, mobile handsets and cars, to name a few. Although this effort ideally supports FAF of all *frugal* components of a product due to limited expanse of ensuing network, forestalling in FAF can be implemented partially as shown later. Even blanket incorporation of FAF will only make *frugal* products into better adaptive systems since bulk of current products are not designed as complex systems with suitable individual parts, their numbers and interactions commensurate to appearance of associated features including emergence and self-organization [32–34]. In other words, individual *frugal* products should at a minimum be pseudo-complex systems—which are adaptive and strictly not complex—possessing *frugal* parts, all of which can adapt, i.e. possessing FAF capability and possibly evolving through successive versions into proper complex systems. Therefore, currently, the bulk of *frugal* products that could be built with blanket FAF can be classified under 'pseudo' due to difficulties of engineering actual complex systems at a local level.

The example of an EV as a *frugal* product is apt due to its ever-growing importance to *sustainable development* [35]. Although a *internal combustion engine* (ICE) vehicle could be viewed as a network of *frugal* components, an EV lends itself to networks, especially due to its simplicity of structure with fewer parts and ability to network with other EVs, relevant entities and a smart grid [36,37]. Besides, EVs are typically low-cost [35] alternatives to ICEs, partly due to possible subsidies, which is

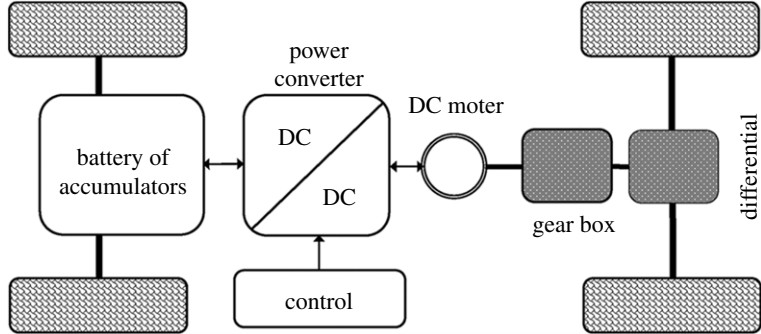

**Figure 2.** Configuration of a battery operated or pure *electric vehicle* [38].

appealing from a *frugal* viewpoint. Therefore, a *frugal* EV built out of *frugal* components could also be utilized for power generation, with a smart grid, other than its primary function serving mobility. Figure 2 shows a general configuration of the parts making up a *battery EV*, which is a pure EV.

## 4.2. Network of frugal products

As ever more *frugal* products become interconnected to each other, a global network spanning multitudes of such products would emerge. Such large-scale interconnectivity would achieve functionality requiring collections of a variety of *frugal* products, which will entail talk among products to forestall failure at individual level including signalling backup products during failures. Although this work assumes inevitability of interconnectivity, the wide variety of *frugal* products from different sectors, such as automotive, aerospace, healthcare etc. should at least talk to each other for sharing information/data for facilitating adaptation through FAF. Such large-scale interaction, hinting at impending dangers, in a large heterogeneous network of *frugal* products possessing inherent FAF abilities coupled with external IoT-connectivity features is key to achieving network complexity. Moreover, reconfiguration of a large network through continuous net addition of adaptive *frugal* products with IoT-connectivity would only make the network larger [39] and enhance its complexity. Successful complex networks of *frugal* products would even approach the vast expanse of the Internet [33] possibly in the distant future.

An instance of a large complex network would be linking of adaptive *frugal* EVs to other such EVs and other relevant entities such as grids—all of them also possessing IoT connectivity—for managing electric-power generation. It should be noted that the assumption of the entire network being *frugal* is relaxed here with only EVs being *frugal* and other entities such as grids and utilities being conventional. Such large-scale interconnectivity would result in robust functionality of both the network and its elements including *frugal* EVs. Other than managing electric-power generation as a collective goal, such a global network of *frugal* EVs would also facilitate safety in mobility by nudging EVs to adapt in response to failure-related information gathered from other network entities.

## 5. Results and discussion

It should be noted that the *factor of frugality*-based methodology has been reported earlier by Rao [9] as the *frugal design approach*. The *frugal design approach* subsumes the classical *factor of safety* while also quantifying the incorporation of other schemes that further streamline a low $S$ design. The current effort uses existing *structural health monitoring* techniques in a new fortification framework to continuously monitor the weaker portion(s) of a *frugal* product designed through the *frugal design approach*. In doing so, sensor(s) are placed at locations deemed weak during design to monitor continuously and hence prevent any impending failure. Moreover, a sensor qualifying for fortification should primarily have proper functionality whose cost should be the lowest possible when compared to other suitable sensors. Although functionality takes precedence to ensure smooth operation of the *frugal* product, a low-cost sensor commensurate to proper functionality should be selected from others in consideration.

A *frugal* product is isolated and as such cannot use features of global complex networks, such as resilience affording a product extra protection, not least by giving extra time for responding to failures. Hence, a strong solution for making individual *frugal* products formidable would involve

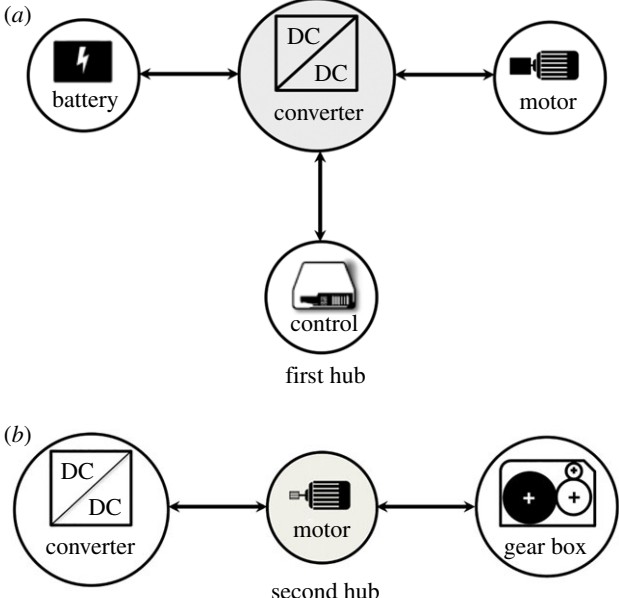

**Figure 3.** Hubs (shown shaded) in the *electric vehicle* of figure 2.

rigorous application of FAF to all their weaker sections. However, generally, such blanket application to a whole product coupled with faster reaction times should be reckoned against focusing on tempering costs. Accordingly, a weak solution would involve partial FAF for certain product topologies that could be pursued as follows. The presence of components that connect to a majority of others, i.e. hubs [33], would make products disassortative [33], where other than fortifying all product components, only hubs should be forestalled through FAF or made to have clones, i.e. backup hubs, depending on the constraint of keeping costs low. A caveat being importance of hubs assumed as solely based on their larger degree or larger numbers of connections when compared to other graphical nodes.

The architecture of a pure EV, shown in figure 2, could also be seen as a local network of *frugal* components. Accordingly, figure 3 shows *power converter* followed by *DC motor* as hubs with their significance in that sequence based on number of connections with other components or nodes. It should be noted that the shaft example given in §3 should be applied for design and fabrication of axles for a *frugal* EV. Other than employing FAF for all components in figure 2, as part of a strong solution, a weaker one would involve fortification of all components but forestalling only hubs of figure 3 and excluding remaining parts from health-monitoring. The degree of cost-reduction together with performance would decide the choice. The 'frugalized' pure EV of figure 2 is a local or isolated pseudo-complex system whose architecture needs to be refined by considering components not shown to arrive at a accurate network representing a successful *frugal* EV.

By contrast, interconnectedness between *frugal* products—themselves pseudo or complete complex systems—in a global complex network would provide an extra layer of protection. In particular, the inherent resilience of a global complex network would facilitate functioning in the face of localized failures and hence further protect *frugal* products that are immediately in vicinity of failure followed by quashing of cascading of failure through remaining products of network. Therefore, blanket FAF of individual products (local networks) together with resilience of a global complex network would increase resistance of individual *frugal* products against failures to a maximum and, thereby, facilitate robust functionality while keeping costs low.

The evolutionary growth of networks of *frugal* products is inevitable due to appealing features of both products and networks, which will be useful as a potent *sustainable* solution against the crises of our time. Therefore, non-engineered growth of a global network of *frugal* products—individually possessing inherent-adaptation and network-integration abilities—aiding *preferential attachment*, because of significance of some products, would nucleate hubs thereby making the network *scale-free* [33], a widely occurring model of small-world phenomenon in complex systems [32]. The emergence of hubs, which are large-degree nodes, will aid accessibility of failure-prone *frugal* products of a dissortative network in a shorter duration of time, which is crucial to the timely control of such networks. Even though growth is

necessary for a successful *scale-free* network, negation of some *frugal* products is essential in the event of failure to prevent a cascade. The widespread morphing and severing activities in a large network of *frugal* products will strengthen or erase connections respectively, thereby making network topology commensurate with robust functionality [40]. In other words, a *frugal* product becomes a dynamic entity whose dynamic interactions with other *frugal* products make the overall network evolve with time. Such dynamic interactions between *frugal* products due to dynamic variation of their states are also a hallmark of complex networks [41].

A possible cost-effective solution to controlling both individual products and their parent dissortative network should involve blanket fortification covering whole network while limiting forestalling to hubs. This strategy is similar to the one applied to local networks of individual products, save for application to entire products instead of components. Consequently, evolutionary growth of such a *scale-free* network would create new hubs and make existing ones larger [33], both of which will play a crucial role in controlling functioning, and hence maintainence of low cost, for the bulk of *frugal* products in the network. Therefore, it is imperative that networks of *frugal* products are allowed autonomous growth to be *scale-free* so that corresponding complex systems have enough constituents for emergence of hubs that relate dissortatively to other elements of their networks.

EVs used for power-generation, besides mobility, are called *gridable electric vehicles* (GEVs) [42], whose *vehicle to house* (V2H), *vehicle to vehicle* (V2V) and *vehicle to grid* (V2G) frameworks for connecting with both home- and community-level grids have been reported in detail by Liu *et al.* [42]. In fact, V2H, V2V and V2G systems inherently possess aggregators, which are controllers coordinating charging/discharging activities between groups of GEVs and grids. Large networks of *frugal* gridable EVs, or *frugal* GEVs, with connections to aggregators in addition to their inherent adaptability, i.e. FAF, coupled with IoT features, will show complex behaviour, especially when used for power generation. As seen in figure 4, where only GEVs are assumed to be *frugal*, the presence of FAF, IoT and GEV controllers will enable appearance of complex behaviour with IoT aiding long-range talk among GEVs and other entities that could be spread across a country, which short-range V2H, V2V and V2G systems cannot penetrate. The long-range IoT wireless connectivity, enabling talk among GEVs, grids, utilities and other entities, will plausibly make its appearance through 5G networks in the foreseeable future [43]. Figure 4 is a simplistic snapshot of a hypothetical scenario that does not show dynamic features including disappearance and appearance of links and/or nodes that bring out behaviour of complex networks. Moreover, the network of figure 4 will have to be scaled up with larger numbers of both links and nodes at a country level involving a national grid system to show complex behaviour.

The *adaptation and networking* (AAN) framework for a given *frugal* case, where adaptation refers to FAF abilities of *frugal* products composing a network, should be tested for its viability at local and global levels. The EV example shows suitability of AAN to a global network of *frugal* GEVs. Also for a given *frugal* product, trade-offs between AAN should be considered to decide whether less rigour can be tolerated through networking so that initial outlays coming from rigour in both design and fabrication can be tempered. Moreover, although this effort assumes all networked products being *frugal*, the AAN framework could be extended to mixed networks of conventional and *frugal* products, as in the example of global network of GEVs. *Artificial Intelligence* (AI) techniques can be used to impart cost-effective complexity to local pseudo systems for learning from product's environment and later using it to facilitate apt interactions between their limited parts. All in all, use of AI could enhance feasibility, and subsequently lead to fruition, of coupling complex-systems theory with *frugal* products to improve functionality at lower costs.

A complex network of *frugal* products will have important ramifications for *sustainable development*. The consumption of less resources in creating individual products would typically entail lesser numbers of activities in various parts of their *life cycles*. This would limit *carbon footprint* to the greatest extent when more and more products become *frugal*. And use of AAN would make the functionality of *frugal* products robust, thereby giving them enhanced longevity and a leg-up vis-à-vis their traditional 'padded' brethren. Moreover, widespread adoption of design for frugality [8,9] would also facilitate effective utilization of EOL systems that could become stranded as newer green or clean technologies start getting increasingly adopted. In fact, products that are currently being streamlined for weight—to achieve less GHG emissions in production and operation—are *frugal* based on the terminology of this effort. Hence, multitudes of *frugal* products can make avail of AAN for giving out more, i.e. efficient functioning, while taking in little, i.e. consuming less resources in their making and also operation.

(*a*)

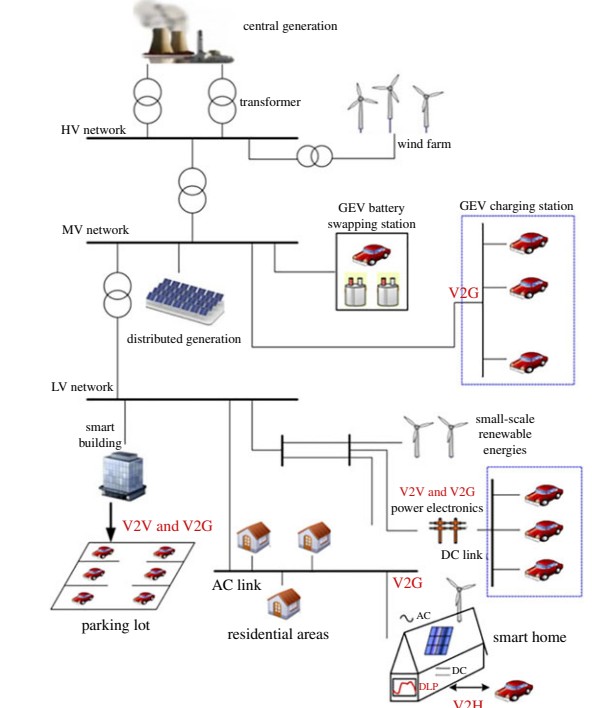

typical integration of V2H, V2V and V2G systems [42].

(*b*)

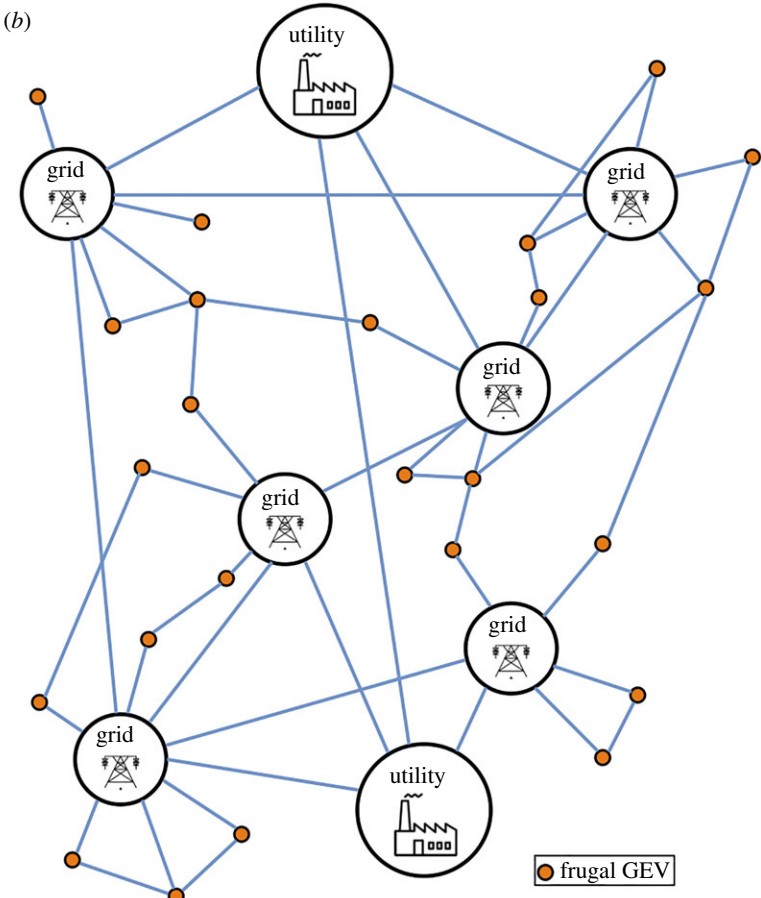

snapshot of a part of large complex network comprising
frugal GEVs and relevant entities for power generation.

**Figure 4.** Complex networking of *frugal gridable electric vehicles*.

# 6. Conclusion

The advent of no-frills *frugal* products built out of constraints on resource consumption is significant for all-round *sustainable development*. This effort has therefore presented an AAN framework to offset vulnerability of *frugal* products to failure, stemming from their structure under even marginal overloading and, thereby realize their potential of simplicity and low cost. Accordingly, complex-systems theory has been invoked to account for increasing numbers of a variety of adaptive *frugal* products getting inevitably interconnected, in the foreseeable future, through IoT-based networks. Critical to the successful implementation of large complex networks of *frugal* products is a two-pronged adaptation methodology developed in this effort for improving product functionality. The adaptation, which is based on rigorous frugal-design approach for 'fortification' and health-monitoring followed by remedial action for 'forestalling', has been exemplified through a *frugal* shaft. Consequently, a cost-effective adaptation strategy for both local and global networks of *frugal* products is blanket fortification of all their constituents with forestalling being limited to hubs. Overall, complex behaviour can be expected from larger heterogeneous networks of interconnected adaptive *frugal* products, as exemplified by a simple example combining GEVs and IoT, with individual products tending in future to local complex systems. As for *frugal* GEVs, the current state of knowledge primarily favours a global complex network of such vehicles for collectively improving the efficacy of managing power generation/distribution with secondary spillover into safety of mobility.

Data accessibility. This article has no additional data.
Competing interests. The author has no competing interests.
Funding. No funding has been received for the article.

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
