## [Reviewer comments · Royal Society Open Science]

Review History

RSOS-192057.R0 (Original submission)

Review form: Reviewer 1 (Massimiliano Ferrara)

Is the manuscript scientifically sound in its present form?

Yes

Are the interpretations and conclusions justified by the results?

Yes

Is the language acceptable?

Yes

Do you have any ethical concerns with this paper?

Yes

Have you any concerns about statistical analyses in this paper?

No

Recommendation?

Accept as is

Comments to the Author(s)

The paper deserves the acceptance in its present form

Review form: Reviewer 2

Is the manuscript scientifically sound in its present form?

Yes

Are the interpretations and conclusions justified by the results?

Yes

Is the language acceptable?

Yes

Do you have any ethical concerns with this paper?

No

Have you any concerns about statistical analyses in this paper?

No

Recommendation?

Accept with minor revision (please list in comments)

Comments to the Author(s)

The paper outlines some design principles for complex frugal products that are composed of networked systems of such products or combination of multiple sub-systems each of which is a frugal product. The purpose is to make the complex frugal product more robust and fail-proof in its functions. The paper is well written and the approach is described clearly. The subject is important enough but a new topic area. The contribution is therefore a welcome one.

a) The two recommendations in this regard are use of structural health monitoring (SHM) of the sub systems, and strengthening (fortification) of the product using design via a factor of frugality approach proposed in an earlier Open Science paper.

Can the author comment on how this approach is any different than conventional design of a non-frugal product, especially the health-monitoring bit? This could be incorporated a bit in the Intro as well as in the Discussion so that the uniqueness of the proposed approach can be better understood.

b) By incorporating structural health monitoring into the product, does the product become less frugal or non-frugal. Ostensibly only certain types of sensors that are cheap may be feasible to incorporate (e.g., RFID devices). Does this restrict the class of products where the SHM approach will work or what types of SHM sensors are feasible under the frugal product definition? Can the author define the attributes of what would qualify as sensors suitable for frugal products?

c) In the shaft example presented, a certain location C is identified in the shaft as being critical. Is this based on the max shear stress theory and prior analysis of the design. Ostensibly, there may be other modes of failure for a product. If so, will such a product require multiple types of SHM implementation? Will this require a priori analysis of all failure modes etc.

d) In the future, I hope the author will present more product case studies where the products are represented by quantitative models of some complexity.

e) There are several places in the manuscript where typos occur - some of these are where "a" be replaced by "an". Also the first line of abstract what does "apt" mean? Is it referring to "appropriate"? This should be spelled out not abbreviated.

Review form: Reviewer 3

Is the manuscript scientifically sound in its present form?

Yes

Are the interpretations and conclusions justified by the results?

Yes

Is the language acceptable?

Yes

Do you have any ethical concerns with this paper?

No

Have you any concerns about statistical analyses in this paper?

No

Recommendation?

Accept with minor revision (please list in comments)

Comments to the Author(s)

1. Take Rao (2013) out on second line, paragraph 1 (page 1). The year of 2013 is too "old", not recent.
2. 3rd line (page 2), change "who" to "that"
3. There are a lot of usage of 'talk' on page 3. I prefer to use "'talk'"
4. "Figure 3.1" on seventh line (page 5): should it be "Figure 1"? Or it relates to Section number?
5. The source of "Figure 3.1"? Ugural (2015)?
6. "Figure 4.1": should it be "Figure 2"? Or it relates to Section number?
7. "Figure 5.1": should it be "Figure 3"? Or it relates to Section number?

Decision letter (RSOS-192057.R0)

Dear Professor Rao

On behalf of the Editors, I am pleased to inform you that your Manuscript RSOS-192057 entitled "ON COMPLEX SYSTEMS OF ADAPTIVE FRUGAL PRODUCTS" has been accepted for publication in Royal Society Open Science subject to minor revision in accordance with the referee suggestions. Please find the referees' comments at the end of this email.

The reviewers and handling editors have recommended publication, but also suggest some minor revisions to your manuscript. Therefore, I invite you to respond to the comments and revise your manuscript.

- Ethics statement

If your study uses humans or animals please include details of the ethical approval received, including the name of the committee that granted approval. For human studies please also detail

whether informed consent was obtained. For field studies on animals please include details of all permissions, licences and/or approvals granted to carry out the fieldwork.

- Data accessibility

If you wish to submit your supporting data or code to Dryad (<http://datadryad.org/>), or modify your current submission to dryad, please use the following link:
<http://datadryad.org/submit?journalID=RSOS&manu=RSOS-192057>

- Competing interests

- Authors' contributions

- Acknowledgements

- Funding statement

Because the schedule for publication is very tight, it is a condition of publication that you submit the revised version of your manuscript before 27-May-2020. Please note that the revision deadline will expire at 00.00am on this date. If you do not think you will be able to meet this date please let me know immediately.

If your manuscript is newly submitted and subsequently accepted for publication, you will be asked to pay the article processing charge, unless you request a waiver and this is approved by Royal Society Publishing. You can find out more about the charges at <https://royalsocietypublishing.org/rsos/charges>. Should you have any queries, please contact opscience@royalsociety.org.

Kind regards,

Anita Kristiansen
Editorial Coordinator

on behalf of R. Kerry Rowe (Subject Editor)
openscience@royalsociety.org

Associate Editor Comments to Author:

Comments to the Author:

The reviewers of your work offer a number of queries and suggestions to improve the manuscript - we would expect you to respond to these and incorporate the changes into the paper. If you choose not to make any of the changes requested, you must provide a full scientific rebuttal for this decision. We look forward to receiving your revised paper in due course.

Reviewer comments to Author:

Reviewer: 1

Comments to the Author(s)

The paper deserves the acceptance in its present form

Reviewer: 2

Comments to the Author(s)

The paper outlines some design principles for complex frugal products that are composed of networked systems of such products or combination of multiple sub-systems each of which is a frugal product. The purpose is to make the complex frugal product more robust and fail-proof in its functions. The paper is well written and the approach is described clearly. The subject is important enough but a new topic area. The contribution is therefore a welcome one.

a) The two recommendations in this regard are use of structural health monitoring (SHM) of the sub systems, and strengthening (fortification) of the product using design via a factor of frugality approach proposed in an earlier Open Science paper.

Can the author comment on how this approach is any different than conventional design of a non-frugal product, especially the health-monitoring bit? This could be incorporated a bit in the Intro as well as in the Discussion so that the uniqueness of the proposed approach can be better understood.

b) By incorporating structural health monitoring into the product, does the product become less frugal or non-frugal. Ostensibly only certain types of sensors that are cheap may be feasible to incorporate (e.g., RFID devices). Does this restrict the class of products where the SHM approach will work or what types of SHM sensors are feasible under the frugal product definition? Can the author define the attributes of what would qualify as sensors suitable for frugal products?

c) In the shaft example presented, a certain location C is identified in the shaft as being critical. Is this based on the max shear stress theory and prior analysis of the design. Ostensibly, there may be other modes of failure for a product. If so, will such a product require multiple types of SHM implementation? Will this require a priori analysis of all failure modes etc.

d) In the future, I hope the author will present more product case studies where the products are represented by quantitative models of some complexity.

e) There are several places in the manuscript where typos occur - some of these are where "a" be replaced by "an". Also the first line of abstract what does "apt" mean? Is it referring to "appropriate"? This should be spelled out not abbreviated.

Reviewer: 3

Comments to the Author(s)

1. Take Rao (2013) out on second line, paragraph 1 (page 1). The year of 2013 is too "old", not recent.
2. 3rd line (page 2), change "who" to "that"
3. There are a lot of usage of 'talk' on page 3. I prefer to use "'talk'"
4. "Figure 3.1" on seventh line (page 5): should it be "Figure 1"? Or it relates to Section number?
5. The source of "Figure 3.1"? Ugural (2015)?
6. "Figure 4.1": should it be "Figure 2"? Or it relates to Section number?
7. "Figure 5.1": should it be "Figure 3"? Or it relates to Section number?

Author's Response to Decision Letter for (RSOS-192057.R0)

See Appendix A.

Decision letter (RSOS-192057.R1)

Dear Professor Rao,

It is a pleasure to accept your manuscript entitled "ON COMPLEX SYSTEMS OF ADAPTIVE FRUGAL PRODUCTS" in its current form for publication in Royal Society Open Science.

You can expect to receive a proof of your article in the near future. Please contact the editorial office (openscience_proofs@royalsociety.org) and the production office (openscience@royalsociety.org) to let us know if you are likely to be away from e-mail contact -- if

you are going to be away, please nominate a co-author (if available) to manage the proofing process, and ensure they are copied into your email to the journal.

on behalf of Prof R. Kerry Rowe (Subject Editor)
openscience@royalsociety.org

Appendix A

Response to Referees

Title: ON COMPLEX SYSTEMS OF ADAPTIVE FRUGAL PRODUCTS

At the outset I would like to thank the reviewers and editors for their positive and valuable comments regarding this research effort. I also thank them for taking the time to point out meticulously issues pertaining to the content of this work. I have accordingly tried to answer the various queries and also revise the initial manuscript. Please find answers to the questions raised by reviewers in the lines that follow. The edited portions appear underlined here and red-colored in the revised manuscript.

Reviewer 1

The paper deserves the acceptance in its present form

ANS: I thank the reviewer for this encouraging and positive comment.

Reviewer 2

The paper outlines some design principles for complex frugal products that are composed of networked systems of such products or combination of multiple sub-systems each of which is a frugal product. The purpose is to make the complex frugal product more robust and fail-proof in its functions. the paper is well written and the approach is described clearly. The subject is important enough but a new topic area. The contribution is therefore a welcome one.

ANS: I thank the reviewer for this encouraging and positive comment.

a) The two recommendations in this regard are use of structural health monitoring (SHM) of the sub systems, and strengthening (fortification) of the

product using design via a factor of frugality approach proposed in an earlier Open Science paper. Can the author comment on how this approach is any different than conventional design of a non-frugal product, especially the health-monitoring bit? This could be incorporated a bit in the Intro as well as in the Discussion so that the uniqueness of the proposed approach can be better understood.

ANS: I thank the reviewer for this query. Here are the reasons for calling it a new *framework* :

1. The existing novel model for *factor of frugality* extends the traditional *factor of safety* and hence offers a new way of designing where material-savings coming from sources in addition to a low *safety factor* are also quantified. This approach also guarantees a rigorous design procedure for achieving robust functionality. In other words, the extension of *safety factor* results in a new design approach whose application to an arbitrary product in an arbitrary sector results in a *frugal* design that is streamlined with a no-frills structure and which is accordingly low-cost.
2. Moreover, by associating with *frugality*, the factor discussed in my effort encourages both *frugal* and non-*frugal* innovators in recognizing the importance of going *frugal* for all round *sustainable development*. In fact, with time designers and engineers would latch on to a *frugal* way of designing and engineering that would subsume even other products which are currently not under the *frugal* banner.
3. The health-monitoring portion of fortification uses existing standard techniques that are applied to a frugally designed part to continuously monitor its weaker sections.

So overall, an extension of classical design based on *factor of frugality*, which is a new concept, is used with standard SHM techniques to fortify frugal products. In other words, what is novel is the combination of an existing novel design approach based specifically on frugality with standard *structural health monitoring* techniques to create a novel *framework* for fortifying frugal products by guarding their weaker sections. I have added the underlined portion below to the discussion section of the revised manuscript.

CONTENT ADDED IN DISCUSSION: It should be noted that the *factor of frugality*-based methodology has been reported earlier by Rao (2019) as the *frugal design approach*. The *frugal design approach* subsumes the classical *factor of safety* while also quantifying the incorporation of other schemes that further streamline a low *S* design. The current effort uses existing *structural health monitoring* techniques in a new fortification-framework to continuously monitor the weaker portion(s) of a *frugal* product designed through the *frugal design approach*. In doing so, sensor(s) are placed at locations deemed weak during design to monitor continuously and hence prevent any impending failure.

b) By incorporating structural health monitoring into the product, does the product become less frugal or non-frugal. Ostensibly only certain types of sensors that are cheap may be feasible to incorporate (e.g., RFID devices). Does this restrict the class of products where the SHM approach will work or what types of SHM sensors are feasible under the frugal product definition? Can the author define the attributes of what would qualify as sensors suitable for frugal products?

ANS: I thank the reviewer for these detailed queries. Here are the answers:

- a) The incorporation of SHM does not change the *frugal* nature of the product. The SHM techniques are only to monitor the sections of the *frugal* product prone to failure.
- b) The sensor cost is an important variable that directly influences the *frugal* nature of the product. So the reviewer is right in pointing to the use of cheap sensors. But cheap here is relative and a sensor with a high cost might be relatively cheaper to other options. Such options have to consider the safety of operation of the *frugal* product and as such the relatively cheapest sensor(s) not comprising on product-functionality should be selected. Accordingly, there is no restriction on the class of products to which this *framework* could be applied.
- c) The sensors that qualify for *frugal* applications include those that are, first and foremost, best in their functionality followed by being relatively cheap when compared to their peers.

CONTENT ADDED IN DISCUSSION: Moreover, a sensor qualifying for fortification should primarily have proper functionality whose cost should be the lowest possible when compared to other suitable sensors. Although functionality takes precedence to ensure smooth operation of the *frugal* product, a low-cost sensor commensurate to proper functionality should be selected from others in consideration.

c) In the shaft example presented, a certain location C is identified in the shaft as being critical. Is this based on the max shear stress theory and prior analysis of the design. Ostensibly, there may be other modes of failure for a product. If so, will such a product require multiple types of SHM implementation? Will this require a priori analysis of all failure modes etc.

ANS: Yes the reviewer is correct in addressing the various modes of failure to be addressed for product fortification. I selected a simple example for brevity and clarity in explaining the various concepts of the new *framework*. The shaft design is based on the *maximum shear stress* theory but other relevant theories including fatigue etc will have to be considered a priori for a real application and the final sensor(s) layout should take all these modes of failure into account.

d) In the future, I hope the author will present more product case studies where the products are represented by quantitative models of some complexity.

ANS: Yes I completely concur with the reviewer. This was the first effort on research laying out the framework. My intent is to apply my *frugal* related research to real time case studies. I am looking forward to indulging in real time applications and there is currently work in progress in this regard in my group.

e) There are several places in the manuscript where typos occur - some of these are where "a" be replaced by "an". Also the first line of abstract what does "apt" mean? Is it referring to "appropriate"? This should be spelled out not abbreviated.

ANS: I have gone through these typos and corrected them wherever I could. The "apt" is just an English word for "proper". I have accordingly replaced this with "a proper" to avoid confusion. I thank the reviewer for pointing to these typos.

Reviewer 3

1. Take Rao (2013) out on second line, paragraph 1 (page 1). The year of 2013 is too “old”, not recent.

ANS: I thank the reviewer for this suggestion but this is a relevant publication and hence I have retained it. I hope the reviewer is not pointing to a formatting error, which I could not locate.

2. 3rd line (page 2), change “who” to “that”

ANS: I have changed to “that” in the required location.

3. There are a lot of usage of ‘talk’ on page 3. I prefer to use “talk”

ANS: I have changed this according to the reviewer’s requirements only on pages 2-3. Subsequent usage does not follow this rule since the reader will get accustomed to the technical context of “talk” in this work.

4. “Figure 3.1” on seventh line (page 5): should it be “Figure 1”? Or it relates to Section number?

ANS: This is actually Figure 3.1. I believe my LaTeX formatting indexes figures according to their sections.

5. The source of “Figure 3.1”? Ugural (2015)?

ANS: I thank the reviewer for pointing what I missed out. I have put the necessary reference in the figure caption.

6. “Figure 4.1”: should it be “Figure 2”? Or it relates to Section number?

ANS: This is again Figure 4.1 in accordance with the template for this manuscript.

7. “Figure 5.1”: should it be “Figure 3”? Or it relates to Section number?

ANS: This is again Figure 5.1 in accordance with the template for this manuscript.

Miscellaneous

Some of the following were already addressed in the initial manuscript and others have also been included below and in the revised manuscript.

Ethics Statement

This research work did not use any animals or humans.

Data accessibility

This work does not have any experimental data.

Competing interests

The author has no competing interests.

Authors' contributions

This is a single author paper with BCR having written the complete manuscript.

Acknowledgements

This work has no individual or entity to acknowledge.

Funding

This work was not supported by any funding agency.